# *PTEN*, *PTENP1*, microRNAs, and ceRNA Networks: Precision Targeting in Cancer Therapeutics

**DOI:** 10.3390/cancers15204954

**Published:** 2023-10-12

**Authors:** Glena Travis, Eileen M. McGowan, Ann M. Simpson, Deborah J. Marsh, Najah T. Nassif

**Affiliations:** 1Cancer Biology, Faculty of Science, School of Life Sciences, University of Technology Sydney, Ultimo, NSW 2007, Australia; glena.travis@alumni.uts.edu.au (G.T.); eileen.mcgowan@uts.edu.au (E.M.M.); 2Central Laboratory, The First Affiliated Hospital of Guangdong Pharmaceutical University, Guangzhou 510080, China; 3Gene Therapy and Translational Molecular Analysis Laboratory, Faculty of Science, School of Life Sciences, University of Technology Sydney, Ultimo, NSW 2007, Australia; ann.simpson@uts.edu.au; 4Translational Oncology Group, Faculty of Science, School of Life Sciences, University of Technology Sydney, Ultimo, NSW 2007, Australia; deborah.marsh@uts.edu.au

**Keywords:** *PTEN*, *PTENP1*, ceRNA networks, microRNAs, precision therapy

## Abstract

**Simple Summary:**

The *PTEN* gene is an important and well-characterised tumour suppressor, known to be altered in many cancer types. Interestingly, the effect of the loss or mutation of *PTEN* is not dichotomous, and small changes in PTEN cellular levels can promote cancer development. Less well-known mechanisms regulating *PTEN,* with emerging importance, include the *PTEN*–miRNA–*PTENP1* axis, which has been shown to play a critical role in the fine tuning of PTEN cellular levels. This mechanism, working at the post-transcriptional level, involves the interplay and competition between the *PTEN* transcript, its pseudogene long non-coding RNA transcripts, *PTENP1*, and microRNAs. Our growing knowledge of this mechanism has opened avenues for the development of strategies to alter the cellular levels of PTEN, miRNAs, and *PTENP1* as a new frontier in cancer therapy.

**Abstract:**

The phosphatase and tensin homolog deleted on chromosome 10 (*PTEN*) is a well characterised tumour suppressor, playing a critical role in the maintenance of fundamental cellular processes including cell proliferation, migration, metabolism, and survival. Subtle decreases in cellular levels of PTEN result in the development and progression of cancer, hence there is tight regulation of the expression, activity, and cellular half-life of PTEN at the transcriptional, post-transcriptional, and post-translational levels. *PTENP1*, the processed pseudogene of *PTEN*, is an important transcriptional and post-transcriptional regulator of *PTEN. PTENP1* expression produces sense and antisense transcripts modulating *PTEN* expression, in conjunction with miRNAs. Due to the high sequence similarity between *PTEN* and the *PTENP1* sense transcript, the transcripts possess common miRNA binding sites with the potential for *PTENP1* to compete for the binding, or ‘sponging’, of miRNAs that would otherwise target the *PTEN* transcript. *PTENP1* therefore acts as a competitive endogenous RNA (ceRNA), competing with *PTEN* for the binding of specific miRNAs to alter the abundance of PTEN. Transcription from the antisense strand produces two functionally independent isoforms (*PTENP1*-*AS-α* and *PTENP1-AS-β*), which can regulate *PTEN* transcription. In this review, we provide an overview of the post-transcriptional regulation of *PTEN* through interaction with its pseudogene, the cellular miRNA milieu and operation of the ceRNA network. Furthermore, its importance in maintaining cellular integrity and how disruption of this *PTEN*–miRNA–*PTENP1* axis may lead to cancer but also provide novel therapeutic opportunities, is discussed. Precision targeting of *PTENP1*-miRNA mediated regulation of *PTEN* may present as a viable alternative therapy.

## 1. Introduction

The phosphatase and tensin homolog deleted on chromosome 10 (*PTEN*), also known as mutated in multiple advanced cancers 1 (MMAC1) and TGFß-regulated and epithelial cell-enriched phosphatase 1 (TEP-1) [1,2,3], is a well-known tumour suppressor gene located on chromosome 10q23.31 [2]. The gene and its protein product play a vital role in cell proliferation, migration, and survival [2,4,5,6,7]. As an antagonist of phosphoinositide 3-kinase (PI3K), PTEN dephosphorylates its substrate PIP_3_ to PIP_2_, thereby negatively regulating the pro-proliferative and anti-apoptotic PI3K/Akt pathway to maintain cellular homeostasis [8,9]. The regulation of PTEN cellular levels is critical in the negative modulation of tumorigenesis with disruption of PTEN signalling leading to significant cellular changes. Interestingly, subtle decreases in cellular levels of PTEN can result in malignancy and tight regulation of the expression, function, and cellular half-life of PTEN, at the transcriptional, post-transcriptional, and post-translational levels is necessary in the prevention of carcinogenesis [10,11]. *PTEN* is frequently mutated and/or deleted in the inherited PTEN hamartoma tumour syndromes (PHTS) [12,13] and multiple sporadic human malignancies, including those from the brain, breast, prostate [1], endometrium [14], skin (melanoma) [15], and colon [6].

Less well-known regulatory mechanisms of *PTEN* with emerging importance include the *PTEN*–miRNA–*PTENP1* axis, which has been shown to play a critical role in the fine tuning of PTEN regulation and cellular integrity. *PTENP1* is a processed pseudogene of *PTEN* termed the phosphatase and tensin homolog pseudogene 1 (*PTENp1*, *PTENpg1*, *PTENP1*, *PTH2*, and *ψPTEN*), which is located on 9p13 (Gene ID: 101243555) [16,17,18]. This pseudogene is transcribed to produce sense and antisense transcripts with the sense transcript showing high sequence similarity with the *PTEN* transcript; however, unlike *PTEN*, this transcript is not translated to produce a protein [19]. Although PTENP1 protein is undetected in cells, when transcribed in vitro as a fusion protein, the product is viable and has comparable phosphatase activity to the wild-type PTEN [19]. The sense and antisense long non-coding RNAs (lncRNA) produced from *PTENP1* are important in the modulation of *PTEN* expression at the transcriptional and post-transcriptional levels, respectively. The *PTENP1* sense transcript (*PTENP1-S*), acting as a competitive endogenous RNA (ceRNA) of *PTEN*, leads to alterations in PTEN cellular abundance. The characteristics of this *PTEN* pseudogene lncRNA include similarities in their microRNA (miRNA) binding sites, and as such, *PTENP1* can act as a decoy or ‘sponge’, competing for miRNAs that target *PTEN*. Disruption of the *PTEN*–miRNA–*PTENP1* axis and ceRNA networks in carcinogenic progression is contemporary and is an exciting area in the discovery of regulatory mechanisms that are altered in cancer. In addition to its regulation of *PTEN* expression, *PTENP1* is able to act as a tumour suppressor independent of its *PTEN* regulatory function as described in a recent review of the role of *PTENP1* in human disorders with a focus on its tumour suppressor functionality [20].

In this review, we outline the importance of *PTEN* regulation in cancer development/progression through the well-known mechanisms of mutation, deletion, and alterations of PTEN structure and function, with a major focus on the role of the *PTEN*–miRNA–*PTENP1* axis. The mechanisms of post-transcriptional regulation of *PTEN*, through interaction with its processed pseudogene (*PTENP1*) transcript (expressed as a lncRNA) and the cellular miRNA milieu, in the context of a cellular ceRNA network is discussed. Knowledge of the working of this regulatory mechanism will allow the identification of potential future novel therapeutic options. Precision targeting of the *PTEN*–miRNA–*PTENP1* axis is important for the regulation of *PTEN* and may present as a viable alternative therapy to increase endogenous wild-type PTEN in tumours shown to have reduced PTEN levels.

## 2. PTEN and Cancer: From Mutations to a Continuum Model of Tumorigenesis

Germline and somatic mutation of *PTEN* is known to contribute to many cancers, highlighting the importance of this tumour suppressor in cancer initiation, progression, and metastasis. Germline mutations of *PTEN* are the cause of four autosomal dominant inherited syndromes: Cowden syndrome (CS) [21], Bannayan–Riley–Ruvalcaba syndrome (BRRS) [22,23], Proteus syndrome (PS), and PS-like syndrome [24], which share common features, including the development of multiple benign hamartomas, and are all classified under the umbrella term of the PTEN hamartoma tumour syndromes (PTHSs) [12,13]. PTHS patients have an increased lifetime risk of developing specific malignancies, mainly breast cancer (approximately 80%) [12,13], thyroid cancer (approximately 30%) [12,13], renal cell carcinoma (approximately 34%) [13], endometrial cancer (approximately 28%) [13], and colorectal cancers (approximately 9%) [13]. In individual PHTS patients exhibiting clinical phenotypes, *PTEN* germline mutations are reported in 25-85% of CS patients [21,25,26], 60% of BRRS [21,22,25,27], up to 20% of PS [28], and between 50 and 67% of PS-like syndrome patients [24]. Interestingly, germline *PTEN* mutations are also associated with a subset of patients with autistic behaviour and extreme macrocephaly [29].

Somatic mutations of *PTEN* are frequently associated with tumorigenesis with somatic alterations of *PTEN* being described in over 50% of cancers of various types [30]. *PTEN* somatic mutations are most prevalent in prostate cancer [31], endometrial cancer [32], melanoma [33,34], non-small-cell lung cancer [35,36], kidney [37], breast cancer [38], and glioblastoma [39]. *PTEN* somatic alterations include the complete loss or inactivation of one allele (functional haploinsufficiency) due to point mutations and/or deletions and/or epigenetic silencing through hypermethylation of the *PTEN* promoter, which is characteristic of some advanced and metastatic cancers [1,4]. Deletion of both alleles of *PTEN* occurs at a lower incidence but is seen mostly in metastatic breast cancer, melanomas, and glioblastomas [1,4,40]. In contrast, a recent study showed that patients with high PTEN expression levels in endometrial cancer had low tumour malignancy, decreased cancer cell proliferation and had a better prognosis [41]. There are different mechanisms of PTEN loss or inactivation, with some being more prevalent in specific tumour types (Table 1) [30,42,43].

The effect of the loss or mutation of *PTEN* is not dichotomous, and subtle changes in PTEN cellular levels have been shown to lead to deleterious consequences relating to tumour incidence, penetrance, and aggressiveness in several epithelial cancers [11,78]. In the hypomorphic transgenic *Pten* mouse, it has been shown that in susceptible organs such as the prostate, PTEN protein expression levels need to reach dramatically low levels (reduced by 70% compared to normal levels) to initiate tumorigenesis, however, in the mammary glands, a more subtle reduction (reduced by 20% compared to normal levels) can initiate tumorigenesis [78]. Thus, *PTEN* does not follow the ‘two-hit’ paradigm or stepwise model of tumour suppressor gene function but rather presents a new continuum model of tumorigenesis whereby tumorigenesis occurs in an incremental dose-dependent manner [11,78]. This has been evidenced in gastric cancer, where *PTEN* expression was shown to gradually decrease with increasing gastric cancer progression [79].

### PTEN Loss, Tumour Immune Evasion, and Therapy Resistance

There are several recent studies that have explored the relationship between PTEN loss and tumour immunity, showing PTEN loss contributes to alterations in the tumour microenvironment (TME) to produce an immunosuppressive niche. The evidence suggests that PI3K signalling may influence the composition and functionality of the TME, thereby modulating the immune response in cancer. Vidotto et al. (2023) analysed PTEN copy number in 9793 cases from 30 tumour types, derived from the Cancer Genome Atlas, and showed that reduced tumour *PTEN* expression occurs with hemizygous loss leading to tumour anti-cancer immune responses [80]. In another integrative analysis of TCGA samples, Lin et al. (2021) found that both PTEN loss and activation of the PI3K pathway were associated with reduced T-cell infiltration and an enhanced immunosuppressive status in multiple tumour types [81]. Overall, the effect of PTEN loss of function in the different cellular compartments swings the balance towards an immunosuppressive TME [82]. There was also a correlation between PTEN loss and poor response to immunotherapy [81]. Interestingly, PTEN loss has also been shown to promote resistance to therapy in breast cancer. Reducing PTEN levels in breast cancer cells conferred resistance to trastuzamab, and patients with PTEN-deficient breast cancers showed poorer therapeutic responses with this drug. Thus, PTEN deficiency has become a good predictor for trastuzumab resistance [83,84]. Reduced *PTEN* expression has been shown in vivo, in mouse models, to be due to specific miRNAs. An example being PTEN as a target of mi-R22 in breast and prostate cancers, which have been shown to have a strong influence in a cancer immune TME, playing a role in cancer initiation, progression, and metastasis [85]. Importantly, in vivo, knockdown of miR-22 appears to invoke tumour resistance in an immunocompetent environment [85]. These findings open new avenues for immuno-targeting, such as modulating miRNAs targeting PTEN, hence improving the efficacy of immunotherapy and overcoming therapy resistance.

## 3. Post-Transcriptional Regulation of *PTEN* by microRNAs and Pseudogene lncRNAs

*PTEN* is constitutively expressed in normal cells and, due to its critical role in several cellular processes, is closely regulated at the transcriptional, post-transcriptional. and post-translational levels to modulate expression, activity, and cellular half-life [86]. One of the most contemporary findings in *PTEN* regulation is the post-transcriptional regulation by its pseudogene long non-coding RNA (lncRNA) and microRNA (miRNA) [87,88,89,90]. This *PTEN*–miRNA–*PTENP1* ceRNA network is discussed in detail in the sections below.

### 3.1. microRNAs Regulate PTEN Expression at the Post-Transcriptional Level

MicroRNAs are single-stranded RNAs comprised of 19-23 nucleotides. These small endogenous RNAs bind to complementary regions within the 3’ untranslated region (UTR) of their mRNA targets, whereby perfect complementarity leads to target degradation and imperfect complementarity leads to the suppression of translation [87,88,89,90] and an overall decrease (or increase in some cases) in target mRNA abundance [87,88]. Mature miRNAs known to repress *PTEN* include, but are not limited to, miR-17, miR-19, miR-21, miR-26, and miR-214 [17]. miRNAs can act as either tumour suppressor miRNAs or tumour promoting miRNAs (oncomiRs), depending on their modulating effect on the expression of their target gene(s). For example, miR-130 acts as a promoter of malignancy through the downregulation of *PTEN* expression in bladder cancer [91], invasive breast carcinoma [92], renal cell carcinoma [93], gastric cancer [94], gliomas [95], lung adenocarcinoma [96], and in colon adenocarcinoma [97]. Expression of miR-130 is lower in both non-small cell lung cancer (NSCLC) cell lines and tissues, and miR-130 overexpression results in cell growth inhibition and enhanced cell apoptosis, through increasing PTEN levels in NSCLC, thus miR-130 acts as a tumour suppressor in this context [98]. Classification of miRNAs into oncomiRs or tumour suppressor miRNAs is complicated, as several miRNAs have been shown to act as either tumour suppressors or oncomiRs in different tumour types [99,100,101].

### 3.2. PTENP1: A Processed Pseudogene of PTEN Produces Bidirectional Transcripts

*PTENP1* is a processed pseudogene of *PTEN*, transcription of which produces unique, bidirectional, sense and antisense, transcripts [102]. Transcription from the sense strand produces a sense transcript (*PTENP1-S*), which is a pseudogene lncRNA with high sequence similarity to the *PTEN* transcript. There is 97.8% sequence similarity within the corresponding coding region (CDS) (with only 18 mismatches) of the two transcripts [17,19,103], and the 3’-UTR of the *PTENP1-S* transcript is approximately 1 kb shorter than that of the *PTEN* transcript. Overall, the 5’-UTR, the CDS, and the early sections of the 3’-UTR of *PTEN* and *PTENP1–S* share high sequence similarity (approximately 95%), however, the level of sequence similarity decreases drastically (approximately 50%) towards the later sections and end of the 3’-UTR [17].

Due to the high sequence similarity between the two transcripts, the *PTENP1-S* transcript shares common miRNA binding sites with the *PTEN* mRNA, particularly at the start of the 3’-UTR, resulting in the ability of *PTENP1-S* and *PTEN* to compete for the binding of common miRNAs. It is now well known that *PTENP1-S* acts as a miRNA sponge to protect *PTEN* from translational repression in a competitive manner, resulting in a positive impact on PTEN expression levels. This competition involves the participation of the *PTEN* and *PTENP1-S* transcripts, as well as the miRNAs targeting these transcripts, in a ceRNA network. Experimentally validated miRNAs that have been shown to participate in the *PTEN* and *PTENP1-S* ceRNA network in various cancer types are summarised in Table 2. miR-21 is a common miRNA shown to target *PTEN* and *PTENP1-S* in prostate cancer [17], hepatocellular carcinoma [104], clear cell renal carcinoma [105], and oral squamous cell carcinoma [106]. Gaining an understanding of this ceRNA network presents the possibility for future manipulation of the network in the treatment of cancers to achieve positive therapeutic outcomes, and this is explored further below.

In addition to the sense transcript of *PTENP1*, transcription from the antisense strand produces two functionally independent isoforms, *PTENP1*-antisense-alpha (*PTENP1-AS-α*) and *PTENP1*-antisense-beta (*PTENP1-AS-β*) [102]. Both isoforms are expressed from convergent promoters and share a *cis* overlap with the *PTENP1-S* transcript and the 5’-UTR of *PTEN* [102,117,118]. The *PTENP1-AS-α* isoform shares high sequence similarity with the 5’-UTR of *PTEN* and is most abundant in the nucleus [102,117,118]. Thus, *PTENP1-AS-α* binds to the 5’-UTR of *PTEN*-associated transcripts, which allows for the localisation of *PTENP1-AS-α* to the *PTEN* promoter region and, in turn, recruits epigenetic modifiers, including chromatin remodelling proteins EZH2 and DNMT3A, which induce the H3K27me3 post-translational histone modification at the *PTEN* promoter, consequently leading to the negative transcriptional regulation of *PTEN* expression [117,118]. The *PTENP1-AS-β* transcript binds to the *PTENP1-S* transcript, which lacks a poly-A tail and provides stability to the *PTENP1-S* transcript through the formation of a *PTENP1-S* and *PTENP1-AS-β* complex that is exported into the cytoplasm, where *PTENP1-S* acts as a miRNA sponge to post-transcriptionally regulate *PTEN* through participation in the ceRNA network [102,117,118] (Figure 1).

## 4. *PTEN*, miRNA, *PTENP1,* and the Endogenous Competitive RNA (ceRNA) Binding Hypothesis

The endogenous competitive RNA (ceRNA) binding hypothesis, first postulated by Pandolfi and colleagues, states that endogenous RNAs, including mRNAs, transcribed pseudogenes, protein-coding genes, lncRNAs, and circular RNAs, compete to regulate each other through binding or sponging of shared miRNAs from the same cellular miRNA pool [119,120]. In this context, *PTEN* has been shown to be regulated by the *PTENP1* sense transcript as *PTENP1-S* acts as a decoy to sequester miRNAs that would otherwise target and repress *PTEN* mRNA translation, thus maintaining or restoring PTEN protein levels [17]. This paradigm challenges previous ideas of sequence conservation working solely to influence the regulation of gene targets by ncRNAs and introduces an additional layer of complexity to the *PTEN* and *PTENP1* ceRNA regulatory network.

Despite the myriad of papers confirming the involvement of *PTEN* and *PTENP1* in a ceRNA network in cervical cancer [115,121], breast cancer [109,111], gastric cancer [116,122], oral squamous cell carcinoma [106], clear cell renal cell carcinoma [105], and in bladder cancer cells [112], there has been controversy from validation consortiums conducting replication studies [123,124]. Such controversy has, however, been recently cleared somewhat by evidence of *PTEN* and *PTENP1* functioning as ceRNAs in studies using CRISPR knockdown and silencing of *PTEN* and *PTENP1* in DU145 prostate cancer cells [125]. *PTENP1* knockdown resulted in the repression of *PTEN* expression [125]. Additionally, the silencing of *PTENP1-AS alpha* and *beta* isoforms resulted in the downregulation of both *PTENP1* and *PTEN* [125], confirming the regulation of *PTEN* and *PTENP1* by the anti-sense isoforms [102]. Further to this, the knockdown of *PTEN* and *PTENP1* resulted in the repression of the *PTENP1-AS* transcripts, while the opposite effect was seen when *PTENP1* was upregulated [125]. This was further evidence for the involvement of *PTEN* and *PTENP1* in a ceRNA network through the mutual regulation of each other’s expression levels [125].

### Other ceRNAs Regulating PTEN Expression in Cancer

While *PTEN* is regulated by its first identified ceRNA, *PTENP1,* other ceRNAs in the form of lncRNAs and proteins that regulate *PTEN*, have since been identified [17]. A number of lncRNAs regulate *PTEN* through ceRNA mechanisms, which include, but are not limited to, *FER1L4* competing with *PTEN* for miR-106a [126] in gastric cancer and miR-18a-5p in osteosarcoma, resulting in the suppression of PTEN [127]. The lncRNA *MEG* interacts with PTEN in a ceRNA manner to bind to miR-19a in glioma [128]. *PTEN* has been shown to be downregulated by the lncRNA *HOTAIR* through interaction with miR-29b in laryngeal squamous cell carcinoma [129]. Additional lncRNAs regulating *PTEN* through ceRNA mechanisms include *Linc-USP16*, which competes for miR-21 in hepatocellular carcinoma, with *PTEN* expression increasing upon its overexpression [130,131]. CASC2 is another proposed coregulator of *PTEN* through a ceRNA regulatory mechanism as *PTEN* and CAS2 both possess miR-21 binding sites [130,132,133]. Other lncRNAs reported to regulate *PTEN* through the ceRNA mechanism include *LINC00702* [134], *NEAT1* [135], *RP11-79H23.3* [136], *TP73-AS1* [137], and *ORLNC1* [138]. Additionally, bioinformatics approaches have identified many other genes as *PTEN* ceRNA competitors, including *TNRC6B*, *RB1*, *TP53*, *NRAS*, *KLF6*, *HIF1A*, *HIAT1*, *CTBP2*, and *TNKS2* [139,140,141]; however, these are yet to be experimentally validated.

## 5. Expression of *PTEN* and *PTENP1* in Cancer

*PTENP1* copy number loss and decreased *PTENP1* expression have been reported in conjunction with *PTEN* loss and decreased expression in several cancer types as the result of either deletion or silencing due to promoter hypermethylation. The *PTENP1* promoter has been shown to be hypermethylated in pancreatic adenocarcinoma, breast cancer, cervical cancer, ovarian cancer, and hepatocellular carcinoma cell lines [142], as well as in lymphoma [143], colorectal cancer [143], clear cell renal carcinoma cells [105,143], and NSCLC tissues [69]. In breast cancer cell lines expressing PTEN protein, *PTENP1* was found to be methylated in MDA-MB-231 cells but unmethylated in MCF-7 cells [144]. *PTENP1-S* was methylated in endometrial cancer and hyperplasia but not in normal tissue [145]. Interestingly, a recent study showed that methylation of *PTENP1* elevated *PTENP1-S* expression in normal endometrium tissue and endometrial hyperplasia from women aged 45 and over, and/or women approaching, or in, menopause [142,146] (Table 3).

*PTENP1* is lost in several cancers and is known to be under selective pressure to undergo copy number loss in cancer. *PTENP1* is lost in melanoma, breast cancer, sporadic colon cancers [17], and in endometrioid endometrial carcinoma [114,147]. Additionally, the low *PTENP1* expression in endometrioid endometrial carcinoma and leukemia cells was shown to be associated with genomic copy number loss of *PTENP1* [148]. In head and neck squamous cell carcinoma cell lines, complete and partial losses of *PTENP1* are known to be frequent; however, the deletion of genomic *PTEN* is not common, further providing evidence for *PTENP1* being under selective pressure to undergo copy number loss in cancer [149]. Furthermore, in studies showing lower levels of *PTENP1* in cancer, it has been predicted to be a promising candidate as a future prognostic biomarker [150].

Generally, *PTENP1-S* expression levels are low compared to *PTEN*. However, the levels of *PTENP1* vary depending on the cell lines and tissues being tested. Some studies have been completed, and the results of the expression of *PTENP1* transcripts relative to *PTEN*, and between the *PTENP1* transcripts in various cell lines and tissues, are presented in Table 4. A quantitative study carried out in a limited number of cell lines showed that the *PTENP1-AS* transcript was more highly expressed compared to the *PTENP1-S* transcript [102]. Additionally, increased expression of the *PTENP1-AS* transcript resulted in lowered PTEN cellular levels due to transcriptional downregulation of *PTEN* by the *PTENP1-AS* transcript [102]. A recent study in melanoma cells showed that increased expression of *PTENP1-AS* resulted in the induction of BRAF inhibitor resistant cells and is likely to be due to the recruitment of epigenetic modifiers to the *PTEN* promoter region, resulting in reduced *PTEN* expression [151]. Furthermore, high levels of *PTENP1-AS* in stage III melanoma patient samples correlated with poor patient survival [151]. To date, all *PTEN* and *PTENP1* expression studies have utilised relative quantitation methods (RT-qPCR) and true cellular levels are yet to be determined. In this context, the use of absolute quantitation methodologies would allow determination of the true cellular levels of *PTEN*, *PTENP1-S,* and *PTENP1-AS* transcripts in normal and cancer cells to help provide an understanding of the perturbations of these levels in cancer cells of various types. This information would be of great importance for our understanding of the contributions of these transcripts in cancer development and progression and would help form the basis of potential future transcript ratio altering therapies for cancer treatment.

### 5.1. PTENP1 also Functions Independently of the PTEN ceRNA Network

Interestingly, the *PTENP1-S* transcript is capable of functioning as a tumour suppressor independently of its *PTEN* regulatory effects and has been shown to have a growth suppressive role in numerous cancers including prostate [17], gastric carcinomas [116], clear cell renal carcinomas [105], gliomas [153], hepatocellular carcinoma [108], bladder [111,112,154], breast [109,110,155], cervical [115], melanoma [156], and colon cancer [17]. In these cancers, overexpression of *PTENP1* led to a decrease in cell proliferation, suppression of cell migration and invasion, and induction of apoptosis through downregulation of the AKT and MAPK signalling pathways as well as downregulation of critical cell cycle proteins cyclin A2 and CDK2, in breast cancer [157]. This further validates the use of *PTENP1* expression levels as a potential future candidate prognostic biomarker due to its tumour suppressor activity. A systematic review and meta-analysis of the data from the literature has recently been published revealing the prognostic value of *PTENP1* expression in cancer. Dai et al. indicate that low expression of *PTENP1* might predict poor prognosis for various carcinomas [150].

Additionally, *PTENP1* has been shown to regulate *PTEN* outside the context of cancer. In smooth muscle cells, *PTENP1* inhibits proliferation and enhances apoptosis [158]. In spinal cord injury, *PTENP1* expression has also been shown to affect recovery by modulating the levels of miR-19b and miR-21 [159]. Interestingly, *PTENP1* expression has implications for fertility as it was shown to regulate the human endometrial epithelial adhesive capacity in vitro by regulating miR-590-3p, and *PTENP1* was shown to be one of a set of highly expressed lncRNAs in human endometrial epithelial cells subjected to blastocyst conditioned medium [160].

### 5.2. Evolution of PTENP1 and cross Regulation of PTENP1 by PTEN

The importance of this functional pseudogene and its regulation of *PTEN* is shown in its evolutionary history and conservation. Tang et al. (2016) have reported the identification of 37 *PTEN* pseudogenes (*PTENPs*) in 65 mammalian genomes, predominantly in primates and rodents [161]. While some *PTENPs* were shared among primates and rodents, others were shown to be species-specific. Of interest, these authors reported the presence of 17 copies of *PTENPs* in the naked mole rat, an anticancer model organism, with all genes sharing common miRNA binding sites with their *PTEN* counterpart [161]. While regulation of PTEN by *PTENP1* is well established, a recent report has shown regulation of *PTENP1* by PTEN in glioblastoma cells [162]. Using targeted methylation and demethylation of the *PTENP1* CpG island, Kovalenko et al. showed that DNA methylation increases *PTENP1-S* expression in the presence of WT-PTEN protein but decreases *PTENP1-S* expression if there is an absence of PTEN protein. They further demonstrated that the PTEN protein binds to the promoter region of *PTENP1* and inhibits *PTENP1-S* expression if its CpG island is demethylated. Thus, in glioblastoma cells, *PTENP1* is a downstream target of PTEN.

## 6. Overexpression of *PTENP1* or Its 3’-UTR: Prelude to Cancer Therapy?

Overexpression of the *PTENP1* 3’-UTR resulted in sequestration of miRNAs, showing *PTENP1* to be an important gene in the regulation of *PTEN*. Overexpression of the 3’-UTR of *PTENP1* in cell lines and in vivo studies has been shown to upregulate PTEN, thereby blocking the PI3K/Akt pathway and decreasing cell proliferation and metastasis, and increasing apoptosis in prostate (DU145) [17], renal (ACHN and SN12MP6) [105], liver (SK-Hep1 and SMMC-7721) [108], breast (MCF-7 and MDA-MB-231) [110,111,157], bladder (T24 and T5637) [112], gastric (MGC803 and BGC823) [116], oesophageal (Eca19), cervical (CasKi and HeLa) [115], and endometrial (RL-952, JEC and HEC-1B) [114] cancer cell lines. However, overexpression of *PTENP1* did not successfully restore PTEN to normal levels in head and neck squamous cell carcinoma cell lines, HN13 and HN30 [149]. Additionally, overexpression of *PTENP1* in oesophageal squamous cell carcinoma cells led to increased *PTEN* levels in Eca19 cells but not in TE-1 cells [163]. In the case of breast cancer, however, it was shown that the levels of PTEN are governed by the estrogen receptor (ER) status of the cells [144]. When *PTENP1* was overexpressed in ER-positive breast cancer cells (MCF-7 and T-47D), *PTEN* expression decreased, and tumour growth was reported to be accelerated in MCF-7 cells [144]. Contrastingly, overexpression of *PTENP1* in ER-negative breast cancer cells (MDA-MB-231 and C3HBA), led to increased *PTEN* expression and inhibition of tumour progression [109,110,111,144]. Similarly, another study in endometrioid carcinoma cells showed that an increase in miR-200c increased estrogen, resulting in an observed decrease in *PTEN* and *PTENP1* expression in cells [114]. Therefore, estrogen plays an essential role in the occurrence of endometrioid carcinoma and affects the negative feedback loop of *PTEN*-miR-200c-*PTENP1* [114]. All the above studies assess the implications of one or two miRNAs targeting *PTEN* and *PTENP1*, neglecting the multiplicity of miRNAs that are able to target and act as either tumour suppressors or oncomiRs, depending on the cell/tissue type in which they are present [101]. An examination of the miRNA expression profiles in a cell/tissue-specific manner will aid our understanding of the miRNAs that are positively and negatively regulated in cancer cells and potentially influencing the *PTEN*–*PTENP1* ceRNA network.

## 7. Manipulating *PTEN*, *PTENP1,* and miRNA Levels as Potential Cancer Therapies

### 7.1. Increasing PTEN Levels Directly

The restoration of functional PTEN has been difficult; however, *PTEN* mRNA levels have been shown to be restored in *PTEN* null cells both in vitro in prostate cancer cells and in vivo in mouse models of prostate cancer through the use of nanoparticles delivering *PTEN* mRNA [164]. Additionally, the restoration of PTEN in *PTEN* null cells resulted in inhibition of the PI3K–Akt pathway and also increased apoptosis. This work represents a new approach to PI3K–Akt pathway inhibition through the restoration of *PTEN* mRNA. Recently, *PTEN* mRNA was delivered via nanoparticles to restore PTEN levels and enhance anti-tumour immunity in melanoma and prostate cancer mouse models [165]. Furthermore, prostate cancer progression has been shown to be inhibited in mice and in a subcutaneous tumour xenograft mouse model by the intraprostatic and intertumoral injection of recombinant adeno-associated virus 9 expressing PTEN [166].

### 7.2. Increasing PTEN Levels Indirectly via PTENP1 as an miRNA Competitor

Transfection of baculovirus packaged *PTENP1* into hepatocellular carcinoma (HCC) cells resulted in increased *PTENP1* levels in cells [107]. The injection of the *PTENP1* expressing baculoviral vector into mice with HCC tumours also reduced tumour growth and cell proliferation, induced apoptosis and autophagy, and inhibited HCC cell properties [107]. Additionally, exosomal *PTENP1* has been transferred from normal cells to bladder cancer cells, which resulted in a reduction in the progression of bladder cancer in vitro and in vivo [154]. *PTENP1* packaged into exosomes has also been transferred to U87MG glioblastoma cells to sponge miR-10a-5p and stabilise PTEN levels in a competitive manner [113]. The success of altering PTEN levels through the delivery of *PTEN* mRNA or *PTENP1* in both in vitro and in vivo studies is a promising start for these candidates for future gene therapies in clinical trials (Figure 2).

### 7.3. Altering Levels of miRNAs Targeting PTEN and PTENP1

miRNAs are powerful gene regulators and are emerging as promising therapeutics in various diseases [167]. Targeting *PTEN* and *PTENP1* with microRNAs has tremendous potential in cancer therapeutics. Increases or decreases in specific microRNAs can lead to an increase or decrease in the levels of *PTEN* and *PTENP1*. For example, in the development of bladder cancer, miR-107 has been shown to be sponged by the LncRNA RP11-79h23.3 in a ceRNA manner to positively regulate *PTEN* expression [136]. In endometrial cancer development, lncRNA LA16C-313D11.11 acts as a ceRNA in the miR-205-5p–*PTEN* axis by inhibiting miR-205-5p and thereby increasing the expression of *PTEN* [168]. lncRNA GAS5 modulates miR-21 in NSCLC cells by increasing PTEN expression [169]. In NSCLC, lncRNA FER1L4 partakes in the positive regulation of *PTEN* by inhibiting cell proliferation and promoting apoptosis in NSCLC [170]. In breast cancer, *PTEN* is influenced positively by the increased expression of *PTENP1* and decreasing miR-20a levels [111]. miR-200 is known to target *PTEN*, a key suppressor of the PI3K/AKT pathway [171,172]. miR-200a negatively targets *PTEN* in endometrial cancer [173] and oesophageal carcinoma [174]. Additionally, miR-200b targets *PTEN* in endometrial cancer [173], along with miR-200c [114], which similarly targets *PTEN* in head and neck carcinoma [175]. miR-429 targets *PTEN* in NSCLC [176]. Thus, in a cancer-dependent context, lncRNAs may be overexpressed to either inhibit or decrease the level of *PTEN* targeting microRNAs, thereby increasing *PTEN* expression and activity as a tumour suppressor. Furthermore, miRNAs can be depleted using miRNA inhibitors, or ‘sponges’ [177], in order to increase *PTEN* or *PTENP1* levels in cells.

On the other hand, microRNAs are known to act as tumour suppressors, whereby they increase the levels of PTEN. A study in endometrioid endometrial carcinoma cells showed that an increase in miR-200c increased estrogen, resulting in a decrease in *PTEN* and *PTENP1* expression in cells [114]. Estrogen plays an essential role in the occurrence of endometrioid carcinoma and affects the negative feedback loop of *PTEN*-miR-200c-*PTENP1* [114]. Thus, the upregulation of certain miRNAs using miRNA mimics [177] could be a useful potential cancer therapy depending on the cancer type.

There are various delivery methods for miRNAs, including virus-based, anti-miRNA oligonucleotide delivery systems. The viral-based delivery systems include the use of retroviral, lentiviral, adenoviral, adeno-associated, and bacteriophage-based vectors [178]. The viral-based miRNA delivery systems are highly immunogenic, toxic, and have size limitations, therefore there is a need to introduce non-viral-based methods for the delivery of miRNAs and anti-miRNA oligonucleotides. The methods utilised have involved the use of lipids, polymers, inorganic and extra-cellular vesicle carriers [178]. There are still many challenges with the use of miRNAs in human trials [167] despite the advancements in miRNA delivery systems, and miRNA drugs have yet to reach phase III human trials [177]. For example, in solid tumours such as hepatocellular carcinomas, the trial of the MRX34 drug, which delivers miR-34 via liposomes intravenously to patients, was terminated due to immune-related severe adverse events [179]. There is therefore a need to understand the regulatory mechanism(s) behind the miRNA [178] actions and their effects on PTEN and *PTENP1* in order to use them for a therapeutic advantage.

Alterations in the levels of *PTEN,* microRNAs, and *PTENP1* are a new frontier in cancer therapeutics with the potential to reverse the cancer phenotype by positively manipulating the *PTEN*–microRNA–*PTENP1* axis in favour of a precancerous cellular phenotype. In the future, successful precision therapeutic targeting in human trials delivering miRNAs and/or *PTEN* and *PTENP1* transcripts will have the possibility of treating various cancers.

## 8. Conclusions

Alterations of the cellular levels of PTEN, miRNAs, and *PTENP1* presents a new frontier in cancer therapeutics with the potential to reverse the cancer phenotype through the positive manipulation of the *PTEN*–miRNA–*PTENP1* axis in favour of pre-cancer levels and induce a pre-cancerous cellular phenotype. The importance of PTEN cellular activity and function has been highlighted in the myriad of studies showing the loss of PTEN expression and/or function as the cause of PHTS and many cancers of various tissue origins. Knowledge of these new mechanisms of post-transcriptional regulation of *PTEN* has opened new avenues for development of novel PTEN-restoring cancer therapies through manipulation of the *PTEN*–miRNA–*PTENP1* axis. Whether through the introduction of *PTEN* mRNA, to increase PTEN cellular concentration, increasing or decreasing *PTENP1* expression, and/or altering the level(s) of specific *PTEN*-regulating miRNAs, it is tempting to consider these future therapies that may allow the fine tuning of PTEN cellular levels to achieve and maintain pre-cancerous levels. While manipulating the *PTEN*–miRNA–*PTENP1* axis holds great promise for the future of cancer therapies, our knowledge of the mechanisms of post-transcriptional regulation of *PTEN*, the various competing components, and the complexities of their interactions needs further study to allow this to become a future reality. Notwithstanding this, future successful human trials delivering miRNAs and/or *PTEN* and *PTENP1* transcripts have great potential in precision therapeutic targeting and the treatment of a broad range of PTEN-related malignancies.

## Figures and Tables

**Figure 1 cancers-15-04954-f001:**
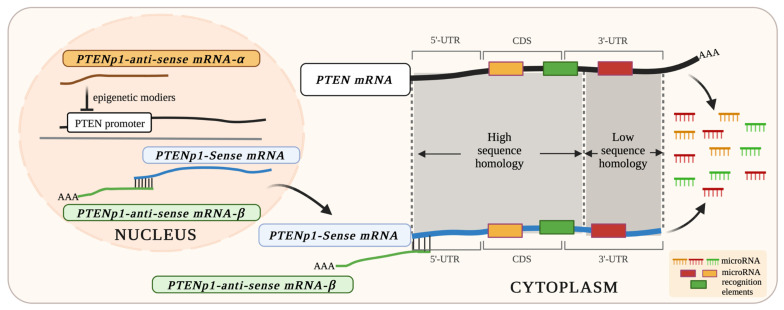
The multifaceted roles of the *PTENP1-S sense* transcript and the two isoforms of the *PTENP1 antisense* transcript (*PTENP1-AS-α* and *PTENP1-AS-β*) in the transcriptional and post-transcriptional regulation of *PTEN* expression. *PTENP1-AS-α* binds to the 5’-UTR of *PTEN*-associated transcripts and localises to the *PTEN* promoter region, where epigenetic modifiers are recruited, resulting in the transcriptional repression of *PTEN*. The *PTENP1-AS-β* transcript binds to the *PTENP1 sense* transcript, which lacks a poly-A tail, and provides stability to this transcript. The *PTENP1-*sense and *PTENP1-AS-β* transcripts form a complex that is exported into the cytoplasm, allowing the *PTENP1 sense* transcript to act as a miRNA sponge to post-transcriptionally regulate *PTEN* (due to the high sequence similarity of the two transcripts) through participation in the ceRNA network (created with Biorender.Com).

**Figure 2 cancers-15-04954-f002:**
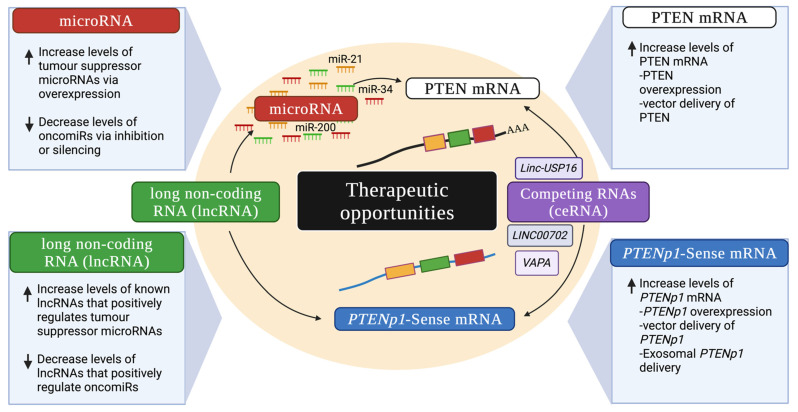
Cancer therapeutic opportunities to restore PTEN levels through the manipulation of *PTEN* mRNA, *PTENP1*, miRNAs, and long non-coding RNAs. MicroRNAs can be therapeutic targets in cancer by increasing or decreasing (shown by the ↑ and ↓arrows, respectively) the levels of either the tumour suppressor microRNAs or oncomiRs, respectively. *PTEN* mRNA levels can be increased through overexpression or the delivery of *PTEN* mRNA into cells to bring the level to a precancerous level and reverse the cancer phenotype. Increasing the levels of *PTENP1-S* through overexpression after delivery into cancer cells leads to ‘sponging’ of miRNAs that would normally bind and repress *PTEN*, leading to increased PTEN levels and reversal of the cancer phenotype. Furthermore, increasing or decreasing the levels of other known lncRNAs that participate in the *PTEN*–miRNA–*PTENP1* ceRNA network to positively modulate tumour suppressor miRNAs or negatively modulate oncomiRs is another approach as a cancer therapeutic (created with BioRender.com).

**Table 1 cancers-15-04954-t001:** Mechanism and frequency (%) of *PTEN* loss in various cancer types.

Cancer Type	Mutation		Deletion		Loss of Protein		Promoter Methylation	
Glioblastoma	30%	[2,3,42,44,45,46]	78%	[44,45,46,47]	65%	[48]	6%	[49]
Breast	3%	[42,50,51]	27%	[38,52]	40%	[42]	35%	[53,54]
Prostate	13%	[55,56,57,58]	51%	[56,57,58]	54%	[55,56,57,58]	<5%	[42,59,60,61]
Colorectal	7%	[6,42,62,63,64,65,66]	8.7%	[42,62,63]	40%	[67]	17%	[68]
Lung	8%	[42]	34%	[42]	56%	[42]	38%	[69]
Endometrial	41%	[14,42,70]	48%	[14,42,70]	45%	[14,41]	19%	[42,71]
Ovarian	16%	[42,43,72,73,74,75,76]	48%	[42,43,72,73,74,75,76]	44%	[42,43,72,73,74,75,76]	10%	[42,77]

Note: Where multiple references are provided, the frequencies of mutation, deletion, and promoter methylation are an approximate average across the relevant publications.

**Table 2 cancers-15-04954-t002:** *PTEN* and *PTENP1*-targeting miRNAs identified and experimentally validated in various cancer types.

Disease	microRNAs (miR) *	References
Prostate cancer	miR-17-5p miR-19-3p miR-21-5p miR-26a-5p miR-214-3p	[17] [17] [17] [17] [17]
Hepatocellular carcinoma	miR-17-5p miR-19b-3p miR-20a-5p miR-193a-3p miR-21	[107] [107] [107] [108] [104]
Clear cell renal carcinoma	miR-21	[105]
Breast cancer	miR-19b miR-20a	[109,110] [111]
Bladder cancer	miR-17	[112]
Glioma	miR-10-5p	[113]
Endometrial cancer	miR-200c	[114]
Cervical cancer	miR-106b	[115]
Gastric cancer	miR-106b miR-93	[116] [116]
Oral squamous cell carcinomas	miR-21-5p	[106]

* All miRNAs target the 3’-UTR of *PTEN* and *PTENP1.*

**Table 3 cancers-15-04954-t003:** *PTENP1* methylation status in various cancer cell lines and cancer tissue types.

Cancer Tissue Type/Cancer Cells	*PTENP1* Promoter Methylation Status	Reference(s)
Breast cancer	Hypermethylated	[142]
MDA-MB-231 breast cancer cells	Hypermethylated	[144]
MCF-7 breast cancer cells	Unmethylated	[144]
Cervical cancer	Hypermethylated	[142]
Ovarian cancer	Hypermethylated	[142]
Hepatocellular carcinoma cell lines	Hypermethylated	[142]
Lymphoma	Hypermethylated	[143]
Colorectal cancer	Hypermethylated	[143]
Clear cell renal carcinoma cells	Hypermethylated	[105,143]
Endometrial cancer and hyperplasia	Hypermethylated	[142,146]

**Table 4 cancers-15-04954-t004:** Expression levels of *PTEN* and *PTENP1* in various cancer types.

*PTEN*:*PTENP1* Relative Expression Ratio *	Cell Line or Tissue Type	References
↑*PTEN*:*PTENP1-S*	Osteosarcoma cell lines	[152]
	Melanoma cell lines	[151]
	Breast cancer cell lines and tissue samples	[102,109,110,111]
	Bladder cancer tissue	[112]
	Gastric cancer cells and tissues	[116]
	Oral squamous cell carcinoma cells	[106]
	Hepatocellular carcinoma cell lines and tissues	[107,108]
	Head and neck squamous cell carcinoma cells	[149]
	Glioma tissue	[112]
	Prostate cell lines	[17]
	Cervical cancer cells	[102,121]
	Endometrioid endometrial carcinoma cells	[114]
	Melanoma	[151]
↓PTEN:*PTENP1-S*	Some prostate cancer tissue samples, gastric cancer cell line, AGS, endometrioid endometrial carcinoma cell lines, RL-952, and JEC	[17] [116] [114]
↓*PTENP1-S*:*PTENP1-AS*	Kidney, HEK-293T, breast, MCF-7, cervix, HeLa, bone, U-2OS	[102] [102] [102] [102]

* Please note: The up and down arrows indicate either an increase or a decrease, respectivly, in the relative expression ratio of the relevant transcripts (*PTEN*, *PTENP1-S* and *PTENP1-AS*) as indicated.

## Data Availability

No new data were created or analysed in this study. Data sharing is not applicable to this article.

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
