# Peer review of "PTEN, PTENP1, microRNAs, and ceRNA Networks: Precision Targeting in Cancer Therapeutics"

_cancers, 2023, doi:10.3390/cancers15204954_

Round 1

Reviewer 1 Report

Travis and colleagues review the role of PTEN pseudogene PTENp1 and the miRNAs and long-noncoding RNAs targeting PTEN in the regulation of PTEN functional network in cells, and their potential as novel anti-cancer therapeutic approaches. The review is well-written and its content is of interest in the field. The Tables and Figures document and illustrate well the different aspects addressed in the text.

Below are some specific minor points

- In page 4, the paragraph from lines 170-175 is confusing and maybe wrong (“In the case of this 5’-UTR” is correct?). Please, revise this.

- It is not clear for this reviewer the use of the nomenclature “PTENp1” instead of “PTENP1”, which is the one more widely used.

- In page 9, lines 115-116, a comment is made on the recent review from Ghafourdi et al on PTENP1 (reference 153). This review should be cited much before in the text, just mentioning recent reviews on the subject.

- It is suggested to cite and briefly comment on the reports by Kovalenko et al (PMID 37619809; an interesting example of PTEN-PTENP1 cross-regulation) and by Tang et al (PMID 27936183; an interesting description of PTEN pseudogenes in mammals).

- Some of the references are cited too many times in the text (usually in a repetitive manner within the same paragraph; see, for instance references 17 and 156). Try to insert the citations at the end of the paragraph, or when the complete description of the findings from the report is made.

- Please, correct some small typos in the text: page 3, lines 104 and 104: “approximetaly”; page 8, line 76: “quanitative”

- In page 10, lines 161 and 164, try to replace one “adittionally” to avoid repetition

Please, see comments to the Authors

Author Response

Comment 1:  In page 4, the paragraph from lines 170-175 is confusing and maybe wrong (“In the case of this 5’-UTR” is correct?). Please, revise this.

Response 1:  The text of this section has been revised accordingly to make this less confusing.

Comment 2:  It is not clear for this reviewer the use of the nomenclature “PTENp1” instead of “PTENP1”, which is the one more widely used.

Response 2: The text has been amended to include PTENP1 throughout the manuscript.

Comment 3:  In page 9, lines 115-116, a comment is made on the recent review from Ghafourdi et al on PTENP1 (reference 153). This review should be cited much before in the text, just mentioning recent reviews on the subject.

Response 3:  The review has been moved and now cited in the introduction section of the manuscript with some additional explanatory information. It is now reference 20.

Comment 4:  It is suggested to cite and briefly comment on the reports by Kovalenko et al (PMID 37619809; an interesting example of PTEN-PTENP1 cross-regulation) and by Tang et al (PMID 27936183).

Response 4:  An additional section has been added to the manuscript. Please see section 5.2 Evolution of PTENP1 and cross regulation of PTENP1 by PTEN.

Comment 5:  Some of the references are cited too many times in the text (usually in a repetitive manner within the same paragraph; see, for instance references 17 and 156). Try to insert the citations at the end of the paragraph, or when the complete description of the findings from the report is made.

Response 5: The references have been checked and streamlined throughout as suggested by the reviewer.

Comment 6:  Please, correct some small typos in the text: page 3, lines 104 and 104: “approximetaly”; page 8, line 76: “quanitative”.

Response 6:  The manuscript has been re-checked and all typographical errors have been corrected.

Comment 7:  In page 10, lines 161 and 164, try to replace one “adittionally” to avoid repetition.

Response 7:  This has been completed.

Reviewer 2 Report

General comments:

The aim of this study was to review the role of PTEN in tumour promotion and progression and to review the mechanisms of PTEN expression regulation with specific emphasis on PTEN - miRNA-PTENp1 axis, which has been shown to play a critical role in the fine tuning of PTEN cellular levels. Overall, the manuscript is clear and flows well and I recommend it for publication but there are several minor points that need to be taken into consideration:

Minor comments:

1.       Table 4 is not clear. Namely, I am not quite sure what is the difference between Ë¥ PTEN:PTENp1-S and Ë¥ PTENp1-S:PTEN. Then, relative expression levels of PTENp1 and PTEN would be much more obvious if the authors put arrows by each gene.

2.       Section 5, subsection 5.1 is about PTENp1 functions and the authors say that “PTENp1-S transcript is capable of functioning as a tumour suppressor independently of its PTEN regulatory effects and has been shown to have a growth suppressive role in numerous cancers...”. It would be desirable to have more details about this important function of PTENp1 with possible mechanisms of action.

3.       Many reports indicate PTEN as responsible for acquired resistance to a given therapeutic protocol and even for the acquisition of multidrug resistance (MDR) to various therapeutic protocols. Especially for breast cancer patients, hormone positive and triple-negative. I think that authors should include this subject in the manuscript.

Author Response

Comment 1:  Table 4 is not clear. Namely, I am not quite sure what is the difference between Ë¥ PTEN:PTENp1-S and Ë¥ PTENp1-S:PTEN. Then, relative expression levels of PTENp1 and PTEN would be much more obvious if the authors put arrows by each gene.

Response 1:  The table has been amended to include relative transcript ratios and arrows have been made consistent.

Comment 2:  Section 5, subsection 5.1 is about PTENp1 functions and the authors say that “PTENp1-Stranscript is capable of functioning as a tumour suppressor independently of its PTEN regulatory effects and has been shown to have a growth suppressive role in numerous cancers...”. It would be desirable to have more details about this important function of PTENp1 with possible mechanisms of action.

Response 2:  Additional information has been added to the paragraph to include aspects of the mechanism of tumour suppression by PTENP1.

Comment 3:  Many reports indicate PTEN as responsible for acquired resistance to a given therapeutic protocol and even for the acquisition of multidrug resistance (MDR) to various therapeutic protocols. Especially for breast cancer patients, hormone positive and triple-negative. I think that authors should include this subject in the manuscript.

Response 3:  An additional section has been added under section 2, which covers aspects of therapy resistance in breast cancer as well effects on anticancer immune functions.

Reviewer 3 Report

While PTEN is a critical tumor suppressor gene involved in controlling cell growth and preventing cancer, its regulation by microRNAs (miRNAs) represents a complex and poorly understood area of research.  A comprehensive understanding of the factors leading to subtle changes in PTEN gene expression in human cancer and inherited PTEN loss syndromes is needed at this time.  This timely review addresses the role of the PTEN- miRNA-PTENp1 axis in cancer.

The review provides recent information on the role of PTENp1, a processed pseudogene of the PTEN gene, in regulating PTEN expression at multiple levels, including transcriptional and post-transcriptional regulation.  The review details how PTENp1 functions as a competitive endogenous RNA, competing with PTEN for the binding of specific microRNAs to modulate the abundance of PTEN in cells. The schematic figures clearly explain the models being discussed.

Furthermore, the review discusses the importance of the PTENp1-PTEN axis in maintaining cellular integrity and how its disruption may contribute to the development and progression of cancer.

The review not only explores the main mechanisms by which PTENp1 influences PTEN expression but also highlights the potential therapeutic opportunities that arise from targeting the PTENp1-miRNA-mediated regulation of PTEN. The final part of the review seeks to provide insights into the molecular mechanisms underlying cancer development related to PTEN dysregulation and proposes precision targeting of PTENp1 as a promising therapeutic strategy for novel cancer treatments.

The review is well-written and presented in a logical way.  However, there is one notable omission. Several recent studies have explored the relationship between PTEN loss and immune functions in tumors, especially using data from large cancer genomics databases like The Cancer Genome Atlas (TCGA) (see citations below). PTEN loss appears to alter the tumor microenvironment, creating an immunosuppressive niche (see three reviews below). This could partly explain why some tumors are resistant to immunotherapy. The modulation of PTEN expression levels could be mediated by miRNA expression.  This additional topic should be discussed as part of section 2 or 5 of their review.

PTEN papers using TCGA

VIDOTTO et al (2023) DOI: 10.1038/s41598-023-31759-6

LIN et al (2021)  DOI: 10.1186/s12885-021-08114-x

Reviews about PTEN and immune response in cancers

CENTOMO et al (2022)  DOI: 10.3390/cancers14246255

CONCIATORI  et al  (2020)  DOI: 10.3390/ijms21155337

VIDOTTO et al (2020) DOI: 10.1038/s41416-020-0834-6

Author Response

COMMENT 1:  The review is well-written and presented in a logical way.  However, there is one notable omission. Several recent studies have explored the relationship between PTEN loss and immune functions in tumors, especially using data from large cancer genomics databases like The Cancer Genome Atlas (TCGA) (see citations below). PTEN loss appears to alter the tumor microenvironment, creating an immunosuppressive niche (see three reviews below). This could partly explain why some tumors are resistant to immunotherapy. The modulation of PTEN expression levels could be mediated by miRNA expression.  This additional topic should be discussed as part of section 2 or 5 of their review.

PTEN papers using TCGA

VIDOTTO et al (2023) DOI: 10.1038/s41598-023-31759-6

LIN et al (2021)  DOI: 10.1186/s12885-021-08114-x

Reviews about PTEN and immune response in cancers

CENTOMO et al (2022)  DOI: 10.3390/cancers14246255

CONCIATORI  et al  (2020)  DOI: 10.3390/ijms21155337

VIDOTTO et al (2020) DOI: 10.1038/s41416-020-0834-6

RESPONSE 1:  An additional section has been added within section 2 to cover these topics (see section 2.1).